# tRNA Derivatives in Multiple Myeloma: Investigation of the Potential Value of a tRNA-Derived Molecular Signature

**DOI:** 10.3390/biomedicines9121811

**Published:** 2021-12-01

**Authors:** Paraskevi Karousi, Aristea-Maria Papanota, Pinelopi I. Artemaki, Christine-Ivy Liacos, Dimitrios Patseas, Nefeli Mavrianou-Koutsoukou, Aikaterini-Anna Liosi, Maria-Anna Kalioraki, Ioannis Ntanasis-Stathopoulos, Maria Gavriatopoulou, Efstathios Kastritis, Meletios-Athanasios Dimopoulos, Andreas Scorilas, Evangelos Terpos, Christos K. Kontos

**Affiliations:** 1Department of Biochemistry and Molecular Biology, Faculty of Biology, National and Kapodistrian University of Athens, 15701 Athens, Greece; pkarousi@biol.uoa.gr (P.K.); partemaki@biol.uoa.gr (P.I.A.); katianaliosi@biol.uoa.gr (A.-A.L.); markalioraki@biol.uoa.gr (M.-A.K.); ascorilas@biol.uoa.gr (A.S.); 2Department of Clinical Therapeutics, School of Medicine, National and Kapodistrian University of Athens, 11528 Athens, Greece; ampapanota@med.uoa.gr (A.-M.P.); liakou@med.uoa.gr (C.-I.L.); dimitrispat@biol.uoa.gr (D.P.); nmavrianou@med.uoa.gr (N.M.-K.); johnntanasis@med.uoa.gr (I.N.-S.); mgavria@med.uoa.gr (M.G.); ekastritis@med.uoa.gr (E.K.); mdimop@med.uoa.gr (M.-A.D.)

**Keywords:** tRNA-derived RNA fragment (tRF), tRNA derivative (tDR), small non-coding RNAs (sncRNAs), hematologic malignancy, plasma cell dyscrasia, bone disease, molecular biomarker, prognosis, post-transcriptional regulator, gene ontology (GO)

## Abstract

Multiple myeloma (MM) is a hematologic malignancy arising from the clonal proliferation of malignant plasma cells. tRNA-derived RNA fragments (tRFs) constitute a class of small non-coding RNAs, deriving from specific enzymatic cleavage of tRNAs. To the best of our knowledge, this is one of few studies to uncover the potential clinical significance of tRFs in MM. Total RNA was extracted from CD138+ plasma cells of MM and smoldering MM patients, and in vitro polyadenylated. First-strand cDNA synthesis was performed, priming from an oligo-dT-adaptor sequence. Next, real-time quantitative PCR (qPCR) assays were developed for the quantification of six tRFs. Biostatistical analysis was performed to assess the results and in silico analysis was conducted to predict the function of one of the tRFs. Our results showed that elevated levels of five out of six tRFs are indicators of favorable prognosis in MM, predicting prolonged overall survival (OS), while two of them constitute potential molecular biomarkers of favorable prognosis in terms of disease progression. Moreover, three tRFs could be used as surrogate prognostic biomarkers along with the R-ISS staging system to predict OS. In conclusion, tRFs show molecular biomarker utility in MM, while their mechanisms of function merit further investigation.

## 1. Introduction

Multiple myeloma (MM) is a hematologic malignancy, belonging to a group of disorders characterized as plasma cell dyscrasias, as it emerges from the clonal proliferation of malignant plasma cells. These malignant plasma cells reside in the bone marrow (BM) and, in the majority of cases, synthesize and secrete a monoclonal immunoglobulin (M-protein) [1]. Monoclonal gammopathy of undetermined significance (MGUS) or smoldering MM (sMM), asymptomatic states characterized by clonal proliferation of malignant plasma cells without symptoms, usually precede MM [2,3]. Symptomatic MM is characterized by the presence of end-organ damage, such as hypercalcemia, renal impairment, anemia, and bone disease, as defined by the acronym “CRAB” [4]. Among these, MM bone disease (MMBD) is a hallmark of MM, as it degenerates dramatically the quality of life of MM patients [5].

The tumor microenvironment plays a catalytic role in MM progression. MM cells interact with cell populations highly abundant in the BM microenvironment, such as mesenchymal stem cells, osteoblast and osteoclast precursors, and immune-system cells leading to cytokine production or cell to cell interactions that promote MM cell proliferation [6]. IL-6 is such a cytokine, which is produced by several cell types in the bone marrow milieu, as well as by MM cells. Moreover, it is characterized as a proliferative factor for MM cells, as it triggers several signaling pathways and also inhibits apoptosis [7,8]. Therefore, the deciphering of the molecular background of MM and MMBD is a necessity for the in-depth understanding of MM and the establishment of new therapeutic options. Regarding MM prognosis, it was until recently based on the International Staging System (ISS), which stratifies patients according to β2-microglobulin and serum albumin levels [9]. In 2015, this staging system was revised (R-ISS), so as to include lactate dehydrogenase (LDH) levels, as well as cytogenetic factors, namely t(4;14) and t(14;16) translocations, and 17p deletion [del(17p)] [10]. However, as MM remains an incurable disease, there is an imperative need for the establishment of novel prognostic biomarkers [11]. The elucidation of the MM molecular background can provide such solutions, as attempts have been made towards the establishment of molecular biomarkers in MM, a typical paradigm of which is the detection of microRNAs (miRNAs) [11,12,13,14].

tRNA derivatives constitute a class of small non-coding RNAs, emerging from explicit enzymatic cleavage of the pre- or mature tRNAs. These fragments have only recently emerged. Therefore, their function has not yet been fully elucidated [15]. Two main classes of tRNA derivatives are currently known: the tRNA halves, or tiRNAs, with a length of 35–50 nt, and tRNA-derived RNA fragments (tRFs), with a length of 15–35 nucleotides (nt). tRFs are further classified into four categories; more specifically, 5′tRFs (also known as tRF-5) incorporate the 5′ of the mature tRNA of origin, while 3′tRFs (also known as tRF-3) incorporate the 3′ of the mature tRNA of origin, and thus are characterized by the presence of the terminal -CCA sequence. tRFs may also derive from the internal part of the mature tRNA, and thus are named “internal tRFs” (i-tRFs). These tRFs might also include the anticodon. Last, tRFs may also derive from the 3′ of the pre-tRNA and thus are named 3′U-tRFs or tRF-1 [16]. Some tRFs have been reported to interact with Argonaute proteins and, similar to miRNAs, participate in the post-transcriptional regulation [17], as well as in the regulation of RNA processing and translation [18]. Moreover, they have been witnessed to exert an important role in neurodegenerative diseases, viral infections, metabolic disorders, inflammation, cancer, and hematologic malignancies [19,20,21]. In cancer and hematologic malignancies, their expression is deregulated; this fact, including their easy detection and high abundance in biological fluids, advocates their usage as molecular biomarkers [22,23]. Members of our research group have attempted to establish such fragments as molecular biomarkers in hematologic malignancies [24,25,26,27].

The present study aimed to further uncover the molecular biomarker utility of tRFs in MM, as only a few studies have elucidated their role in this disease so far [28,29,30]. Thus, we attempted to quantify six tRFs, which were previously identified by members of our research group, in CD138+ plasma cells of sMM and MM patients. We used an innovative in-house-developed quantitative PCR (qPCR) assay and examined their potential value as diagnostic and prognostic biomarkers, by performing extensive biostatistical analysis. Last, we conducted an in silico analysis, to investigate the putative function of one of the tRFs studied.

## 2. Materials and Methods

### 2.1. Sample Collection 

Bone marrow aspirate (BMA) samples were collected from 94 patients. Seventy-six of them were MM patients and 18 of them were diagnosed with sMM. Adult patients with a recent diagnosis of MM or sMM were included in the study, while patients who had already received any kind of treatment or suffered from any other concomitant malignancy were excluded. The median age of MM patients was 68 years (range: 35–88 years), while the median age of sMM patients was 66 years (range: 54–81 years). The presence of high-risk cytogenetic abnormalities, namely del(17p), t(4;14), t(14;16), and (+1q) was assessed using fluorescence in situ hybridization (FISH). The MM patients were classified according to ISS and revised ISS (R-ISS) staging systems, as defined by the International Myeloma Working Group (IMWG) in 2005 and 2015, respectively [9,10]. Whole-body low-dose computed tomography (WBLDCT) was used to assess osteolysis. The characteristics of the MM patients are shown in Table 1.

Only one patient did not receive any treatment, while the treatment of the rest 75 patients varied. More specifically, 63 out of the 75 MM patients (84.0%) were treated with bortezomib plus an immunomodulatory drug (either lenalidomide (60 cases) or thalidomide (3 cases)), 6 (8.0%) MM patients received bortezomib along with cyclophosphamide and dexamethasone, and 6 (8.0%) MM patients were treated with lenalidomide and dexamethasone. Moreover, 21 (27.6%) out of 76 MM patients were subjected to autologous stem cell transplantation following high-dose melphalan (HDM-ASCT), whereas the rest 55 (72.4%) MM patients were not candidates for bone marrow transplant, either because of being older than 65 years (48 cases) or because of severe comorbidities and/or impaired performance status (7 cases).

This study was approved by the Ethics Committee of the “Alexandra” General Hospital of Athens and was conducted according to the principles of the Declaration of Helsinki. Written informed consent was obtained from each participant of this study.

### 2.2. CD138+ Plasma Cell Selection

In order to select plasma cells from BMA samples of MM and SMM patients, we performed CD138-positive selection among mononuclear cells of BMA as a method of choice. CD138, officially known as Syndecan 1, is a transmembrane (type I) heparan sulfate proteoglycan encoded by the human *SDC1* gene; its expression is considered to be a hallmark of MM cells and plasma cells in the bone marrow [31,32,33]. Therefore, 10 mL of BMA were collected in tubes containing ethylenediaminetetraacetic acid (EDTA) and BMA samples were processed, immediately after collection, for CD138 enrichment. More specifically, the Ficoll-Paque technique was used to isolate mononuclear cells from BMA. Following that, a positive selection of CD138+ plasma cells was performed using anti-CD138–coated magnetic beads (Miltenyi Biotec, Bergisch Gladbach, Germany).

### 2.3. Total RNA Extraction, In Vitro Polyadenylation, and cDNA Synthesis

Total RNA was isolated from CD138+ plasma cells, using TRI Reagent^®^ (Molecular Research Center Inc., Cincinnati, OH, USA). The concentration of each total RNA was determined using Qubit™ 2 Fluorometer (Invitrogen™, Thermo Fisher Scientific Inc., Carlsbad, CA, USA), and its integrity was assessed by agarose gel electrophoresis. Next, 200 ng of each total extract were subjected to in vitro polyadenylation, using *E. coli* poly(A) polymerase (New England Biolabs Ltd., Ontario, ON, Canada) and 80 μM ATP. Each sample was incubated at 37 °C for 60 min, followed by an inactivation step at 65 °C for 10 min. Reverse transcription was then performed, using MMLV reverse transcriptase (Invitrogen™, Thermo Fisher Scientific Inc.) and an oligo-dT–adapter primer, following the manufacturer’s instructions. The oligo-dT–adapter primer sequence was 5′-GCGAGCACAGAATTAATACGACTCACTATAGGTTTTTTTTTTTTVN-3′. Both polyadenylation and reverse transcription were conducted in a GeneAmp^®^ PCR System 9700 (Applied Biosystems™, Thermo Fisher Scientific Inc., Foster City, CA, USA).

### 2.4. tRF Selection

Six (6) tRFs, specifically, five (5) i-tRFs, namely i-tRF-Pro^TGG^, i-tRF-Glu^CTC^, i-tRF-His^GTG^, i-tRF-Gly^GCC^, i-tRF-Phe^GAA^, and one (1) 3′tRF, namely 3′-tRF-Leu^AAG/TAG^, were selected for this study. The tRFs investigated in this study are shown in Figure 1. All tRFs have previously been verified experimentally by members of our research group, after bioinformatical analysis based on publicly available RNA sequencing data. MINTbase v.2.0, one of the few databases comprising tRFs detected in human tissues, was used to align the tRFs with the respective tRNAs of origin [34]. The names of the tRFs were determined based on the tRF category, the amino acid carried by the tRNA of origin, and the respective anticodon. The sequences of these tRFs were submitted to GenBank^®^ of NCBI. The fragment sequences, the anticodons, the localization of the tRNAs of origin, their MINTbase unique IDs, as well as their GenBank^®^ accession numbers are shown in Table 2. 

### 2.5. Real-Time qPCR

Specific forward primers were designed for each tRF, *SNORD43* (also known as RNU43) and *SNORD48* (also known as RNU48), which were used as reference genes, due to their similar size with tRFs and stable expression. Each forward primer was used along with a common universal reverse primer, complementary to the oligo-dT–adapter used during cDNA synthesis. All primers used in the qPCR assay are shown in Table 3. The 10-μL reaction mixture was composed of 0.5 μL of 10-fold diluted cDNA, 5 μL KAPA™ SYBR^®^ FAST qPCR master mix (2X) (Kapa Biosystems Inc., Woburn, MA, USA), 1 μL of each primer to a final concentration of 200 nM each, and 2.5 μL RNase-free H_2_O. The qPCR assay was performed in an ABI 7500 Fast Real-Time PCR System (Applied Biosystems™), following a standard thermal protocol for cycling and melting, as previously described [26]. The levels of each tRF were determined using the comparative threshold cycle (2^−ΔΔCt^) method [35,36]. The normalized levels of each tRF were measured in relative quantification units (RQUs), as the ratio of tRF molecules to the geometric mean of *SNORD48* and *SNORD43* molecules, divided by the same ratio that had been previously determined for a cDNA pool, consisting of 5 cDNAs deriving from patient samples, which was used as a calibrator. The qPCR assay was optimized, in order to observe a unique melting curve for each amplicon, and thus be able to quantify the selected tRFs in all samples. The melting curves for each tRF, as well as those of the reference molecules, are shown in Appendix A.

### 2.6. Biostatistical Analysis

Data analysis was performed using SPSS^®^ (v.26) software (IBM Corporation, Armonk, NY, USA). Due to the non-canonical distribution of the levels of each tRF in the patient cohorts, non-parametric tests were conducted. The X-tile algorithm was used to generate optimal cut-off points [37], which were used to classify the patients in distinct subgroups, based on the levels of each tRF. The Mann–Whitney *U* test was used to assess the significance of differences in tRF levels between distinct groups. Putative associations between each tRF expression and other categorical variables were assessed with the chi-square test or Fischer’s exact test. Regarding survival, Kaplan–Meier curves, concerning overall (OS) and progression-free (PFS) survival of MM patients, were built, and differences between the curves were assessed using the Mantel–Cox (log-rank) test [38]. Last, bootstrap (1000 samples) univariate and multivariate Cox regression analyses were carried out, estimating bias-corrected and accelerated (BCa) 95% confidence intervals (CI) of each hazard ratio (HR). Statistical results were considered significant only when the *p* value score was less than or equal to 0.05.

### 2.7. In Silico Analysis for tRF Target Prediction and Gene Ontology (GO) Enrichment Analysis

Two databases, namely tRFTar [39] and tRFtarget [40] were used to predict the target genes of each tRF. tRFTar uses cross-linking immunoprecipitation-sequencing (CLIP-seq) data to provide putative targets, while tRFtarget uses prediction tools. In tRFtarget, the default parameters were used (free energy ≤−10 Kcal, and maximum complementary length >8). Binding sites in the whole transcript were chosen in both databases. However, none of the databases provided information about the targets of the i-tRFs included in this study, but only for 3′-tRF-Leu^AAG/TAG^. The observed common target genes of 3′-tRF-Leu^AAG/TAG^ were used for GO enrichment analysis; this analysis categorized the target genes based on the cellular component they act, the biological process in which they are implicated, and the molecular function they exert and provides an enrichment score, which shows if a set of genes associated with a specific cellular component/biological process/molecular function is over-presented among the others.

## 3. Results

### 3.1. Differences in tRF Levels of CD138+ Plasma Cells between sMM and MM Patients, as Well as among MM Patients’ Subgroups

A mean ± standard error (S.E.) of 181.1 ± 62.5 RQU was observed for 3′-tRF-Leu^AAG/TAG^ levels in sMM patients’ CD138+ plasma cells, while a mean ± S.E. of 51.5 ± 10.5 RQU was observed for the levels of the same tRF in MM patients’ CD138+ plasma cells. As revealed by the Mann–Whitney *U* test, this difference is statistically significant (*p* = 0.041); these results are shown in Figure 2A. The levels of the other five tRFs did not differ significantly between sMM and MM patients.

In order to examine potential relationships between tRFs and MM features, we checked whether the tRF levels differed between distinct subgroups of MM patients, including those with different M-protein isotype as well as patients with or without osteolytic lesions. No statistically significant differences were observed in the levels of each tRF among MM patients with different M-protein isotype (IgG and IgA). On the other hand, the Mann–Whitney *U* test revealed significantly higher levels of i-tRF-Gly^GCC^ in the CD138+ plasma cells of patients without osteolytic lesions (mean ± S.E. = 47.8 ± 18.3), compared to those with osteolytic lesions (mean ± S.E. = 17.7 ± 5.1), as shown in Figure 2B (*p =* 0.047). 

### 3.2. tRFs as Promising Molecular Indicators of Favorable Prognosis in MM

MM patients were subgrouped based on the levels of each tRF, based on the cut-off points determined for prognostic purposes using the X-tile software, as aforementioned. The frequencies of MM patient subgroups are shown in Appendix A. Interestingly, positive i-tRF-Gly^GCC^ status was found to be associated with the presence of chromosomal aberration (+1q) (Appendix A). Associations between each tRF status and del(17p), t(4;14) or t(14;16) could not be checked, due to the limited number of MM patients bearing at least one of these chromosomal aberrations.

The median follow-up time of MM patients was 24 months. During the accrual period, 29 MM patients showed disease progression, while 13 MM-related deaths were also recorded. As revealed by Kaplan–Meier survival analysis, MM patients with high i-tRF-Pro^TGG^, i-tRF-Glu^CTC^, i-tRF-His^GTG^, i-tRF-Phe^GAA^, or 3′-tRF-Leu^AAG/TAG^ levels show significantly prolonged OS, compared to those with lower levels of each of these molecules (*p =* 0.024, 0.014, 0.003, 0.040, and 0.039, respectively). These data are presented in Figure 3. Additionally, i-tRF-Gly^GCC^ and 3′-tRF-Leu^AAG/TAG^ can predict disease progression in MM patients. The PFS intervals were significantly longer in MM patients with high i-tRF-Gly^GCC^ or 3′-tRF-Leu^AAG/TAG^ levels (*p =* 0.016 and 0.013, respectively). These results are presented in Figure 4, while the putative biomarker utility of all tRFs is summarized in Figure 5.

Moreover, patients who underwent HDM-ASCT and had high i-tRF-Pro^TGG^, i-tRF-Glu^CTC^, or i-tRF-His^GTG^ levels show significantly prolonged OS, compared to those who received HDM-ASCT and had lower levels of each of these molecules (*p =* 0.008, 0.045, and 0.019, respectively). Furthermore, patients who did not receive HDM-ASCT and had high i-tRF-His^GTG^ levels showed significantly longer OS intervals compared to patients with low i-tRF-His^GTG^ levels (*p =* 0.033) These results are presented in Appendix A. Regarding PFS, patients who received HDM-ASCT and had high i-tRF-Gly^GCC^ or 3′-tRF-Leu^AAG/TAG^ levels, show significantly prolonged PFS intervals in comparison with patients who received HDM-ASCT and had lower i-tRF-Gly^GCC^ or 3′-tRF-Leu^AAG/TAG^ levels (*p =* 0.039 and 0.022, respectively). These results are presented in Appendix A. 

Intriguingly, bootstrap multivariate Cox regression analysis revealed that the prognostic significance of i-tRF-Pro^TGG^, i-tRF-Glu^CTC^, and i-tRF-His^GTG^ regarding MM patients’ OS is independent of that of R-ISS staging. Moreover, the prognostic significance of i-tRF-Gly^GCC^ and 3′-tRF-Leu^AAG/TAG^ is independent of ISS staging. These results are presented in Table 4.

### 3.3. In Silico Functional Analysis of 3′-tRF-Leu^AAG/TAG^

tRFTar and tRFtarget databases revealed 755 and 1486 target genes for 3′-tRF-Leu^AAG/TAG^, respectively. We preferred to perform the downstream analysis using the common target genes (662) provided by the two databases, as shown in Figure 6. The subsequent GO analysis of this gene list revealed 48 significantly enriched cellular components, 268 significantly enriched biological processes, and 63 significantly enriched molecular functions. Each enrichment score was considered as significant only when the *p* value was less or equal to 0.05. Based on these results, 3′-tRF-Leu^AAG/TAG^ was found to be implicated among others in the regulation of gene expression, WNT signaling pathway, and steroid hormone signaling. In Figure 7, the cellular components, molecular functions, and biological processes showing the highest enrichment scores are graphically presented.

## 4. Discussion

By virtue of small RNA sequencing and other novel approaches, tRNA derivatives represent a study field that although it is still in its infancy, has arisen scientific interest. Their involvement in various molecular and cellular processes, including transcriptional regulation, cell growth, and differentiation, has highlighted their potential utility as biomarkers and therapeutic targets against cancer, as these processes are directly linked to cancer development [41]. Their deregulated levels in cancer have been variously reported, and numerous studies have attempted to associate this deregulation with the malignant state [42]. However, to the best of our knowledge, only a handful of research studies have attempted to investigate the role of tRFs in MM [28,29,30].

Firstly, our study proved the abundance of tRFs in the CD138+ plasma cells of MM and sMM patients. Moreover, we showed that 3′-tRF-Leu^AAG/TAG^ was significantly overexpressed in sMM cases compared to MM patients. Nowadays, many imaging modalities that make bone disease detection extremely exact and differential diagnosis between sMM and MM simple and accurate are available. However, in some ambiguous cases, in which setting a diagnosis is really important [43], there is an unmet need for the discovery of molecular biomarkers able to distinguish between sMM and MM patients. Given that, 3′-tRF-Leu^AAG/TAG^ most likely represents a novel molecular biomarker that could prove useful for the differential diagnosis of MM from sMM. Thus, further research in larger cohorts is required in order to establish such a role of this molecule. In the same context, i-tRF-Gly^GCC^, which is overexpressed in patients without osteolytic lesions, may serve as a biomarker able to distinguish between patients with and without bone disease.

Our study revealed a putative prognostic value in MM for all six molecules studied. More specifically, higher levels of i-tRF-Pro^TGG^, i-tRF-Glu^CTC^, i-tRF-His^GTG^, and i-tRF-Phe^GAA^ were associated with superior OS, higher levels of i-tRF-Gly^GCC^ were associated with prolonged PFS, while higher levels of 3′-tRF-Leu^AAG/TAG^ correlated both with prolonged OS and PFS in the total population. Currently, the R-ISS staging system is universally used as a tool able to predict survival in MM patients. R-ISS staging consists of three subgroups with different survival rates; R-ISS I (5-year OS rate: 82%), R-ISS II (5-year OS rate: 62%), and R-ISS III (5-year OS rate: 40%) [10]. Our study revealed that three of the investigated tRFs, namely i-tRF-Pro^TGG^, i-tRF-Glu^CTC^, and i-tRF-His^GTG^ have a prognostic value independent of R-ISS staging. This means that those molecules may be able to serve in the future as additional prognostic biomarkers in MM that could even be incorporated or combined with R-ISS. It would be interesting to evaluate their prognostic significance and their ability to subclassify R-ISS stage II patients, who represent the larger and most heterogeneous R-ISS class (62% of the total population was classified as R-ISS II in the relevant publication) [10]. Moreover, all tRFs retained or showed a tendency for retaining their prognostic significance in patients who were subjected to HDM-ASCT. In the subgroup of patients not subjected to HDM-ASCT, i-tRF-His^GTG^ retained its prognostic significance regarding OS. Three out of the six tRFs investigated in this study, namely i-tRF-Gly^GCC^, i-tRF-Phe^GAA^, and 3′-tRF-Leu^AAG/TAG^ have already been proposed as biomarkers in CLL by previous works of members of our research group [24,25,27]. tRFs deriving from tRNA^GluCTC^ have been detected in exosomes of cancer patients [44], while i-tRFs deriving from tRNA^ProTGG^ and tRNA^HisGTG^ were shown to have prognostic value in squamous cell carcinoma of the head and neck [45]. These data highlight their implication in cancer progression and their utility as molecular biomarkers.

i-tRFs represent the least investigated tRF class. This may happen due to the fact that the cleavage position for i-tRF emergence varies, generating a large number of i-tRFs, some of which have subtle differences and subsequently, are difficult to be distinguished. A problem deriving from the poor investigation of i-tRFs is that an in silico functional analysis could not be carried out for the five i-tRFs included in this study, due to the lack of CLIP-seq or prediction data [30]. On the contrary, 3′tRFs represent the most well-investigated tRF class. This tRF category shares functional similarities to miRNAs, as they interact with Argonaute (AGO) proteins, and scientific evidence shows that they probably have a seed sequence, located at the same sequence positions as the respective seed sequence of the miRNAs [46]. 

Interesting results were obtained from the functional analysis concerning 3′-tRF-Leu^AAG/TAG^. This tRF was shown to be implicated in processes related to oncogenesis. More specifically, this tRF was found to interact with molecules that exert their function in the podosome and actin cytoskeleton. Podosomes are actin-rich structures anchored to the extracellular matrix [47], which have been shown to drive osteoclast-dependent bone resorption [48], a typical characteristic of MMBD [5]. Another cellular component showing a high enrichment score is the vesicle coat; MM cell-derived extracellular vesicles (EVs) play a significant role in MM proliferation, by providing the BM healthy cell populations with a cargo that facilitates malignant cell proliferation [49,50]. 

Regarding the possible implication of 3′-tRF-Leu^AAG/TAG^ in specific biological processes, macropinocytosis is intriguingly interesting, as this process represents an amino acid supply mechanism for cancer cells [51]. In MM, macropinocytosis is also a mechanism for EV internalization. In MM cells bearing a mutation in *RAS*, a common MM patients’ mutation, this process was also shown to promote MM cell growth and survival, through glutamine provision [52]. 

Other significantly enriched biological processes related to MM are positive regulation of the estrogen receptor pathway and in general steroid hormone receptor pathway regulation. Glucocorticosteroids, one of the first treatments for MM, remain the cornerstone of MM treatment, even in the era of new agents, where glucocorticosteroids, especially dexamethasone, are a part of the majority of anti-myeloma treatment regimens [45,53]. The cytotoxic effect of dexamethasone in MM cells is well known and the mechanism behind that seems to be the upregulation of pro-apoptotic genes and downregulation of anti-apoptotic genes [54,55]. Decreased glucocorticoid receptor expression has been reported in MM patients, leading to resistance to glucocorticoid therapy [56]. Even more interestingly, 3′-tRF-Leu^AAG/TAG^ was found to interact with genes that bind to steroid hormone receptors, which presents the most significantly enriched molecular function. Concerning the estrogen receptor, its activation inhibits IL-6–mediated MM cell growth, by inducing PIAS3 and leading to blockage of STAT3-induced signaling [57]. PIAS3 also blocks RANKL (also known as TNFSF11) expression, leading to inhibition of the RANK (also known as TNFRSF11A) receptor signaling and subsequently to MMBD reduction, as this pathway is directly linked to the genesis of osteolytic lesions [58]. 

Furthermore, the GO analysis showed that 3′-tRF-Leu^AAG/TAG^ contributes to beta-catenin-TCF complex assembly. Beta-catenin (also known as CTNNB1) is the main mediator of WNT signaling; its interaction with the transcription factor TCF enhances the transcription of its target genes and consequently enhances bone formation. Moreover, MM cells secrete WNT inhibitors, leading to aberrant WNT-mediated signaling, an aspect associated with MM pathogenesis [59]. The high enrichment score of this biological process comes in line with the fact that the molecular function “Beta-catenin binding” is one of the molecular functions with the highest enrichment scores. 

Most of the highly enriched molecular functions, combined with the also highly enriched biological process “mRNA transcription by RNA polymerase II”, could be explained by the regulatory role imputed to 3′tRFs, regarding gene expression. These molecular functions include promoter-specific chromatin binding, steroid hormone receptor binding, chromatin DNA binding, translation regulator activity, nucleic acid binding, translation regulator activity, chromatin binding, protein C-terminus binding, transcription coactivator activity, DNA-binding transcription factor binding, and transcription factor binding. All these data indicate that investigation of the mechanistic role of 3′-tRF-Leu^AAG/TAG^ in MM could reveal new aspects, concerning the MM molecular background; however, due to the lack of experimental validation, we cannot be sure about the implication of this fragment in one, some or all of the processes described above. Thus, experimental verification is a necessity to provide a clearer perspective regarding the implication of 3′-tRF-Leu^AAG/TAG^ in specific molecular processes and how these processes are connected.

Our study has some limitations. Firstly, the cohort size is not very large; nevertheless, it is important to mention that our study population is not subjected to age or performance status limitations and is quite indicative of the real-world MM population. Due to the small sample size, our study can only present some evidence about the clinical utility of the aforementioned tRFs as molecular biomarkers in MM. For the same reason, no associations between tRF levels and features of patients with rare MM types, such as IgD and non-secretory myeloma type, could be studied. Moreover, the results regarding the association of positive i-tRF-Gly^GCC^ status with the presence of (+1q) need to be validated in a larger cohort of MM patients. Secondly, the median follow-up time of the patients included in the current study is only 2 years; however, a longer follow-up time is needed to uncover the emerging potential of particular tRFs as important molecular biomarkers in MM. Additionally, a cohort consisting of normal controls is not available, as BMA is a painful procedure with rare but still possible complications, performed only when the presence of particular hematologic diseases is suspected. However, unprocessed bone marrow samples could be purchased and used for comparison purposes. The inclusion of such samples could strengthen our conclusions about the diagnostic and prognostic utility of these tRFs in MM. Moreover, a weakness of our qPCR assays developed for tRF quantification is that fragments with slight sequence differences generated from the same tRNAs of origin are considered as a single one and are hence cumulatively quantified. Lastly, another limitation is that the selected CD138+ plasma cells included not only malignant plasma cells but also normal ones [60]. Furthermore, a subpopulation of malignant plasma cells, which are more primitive and which have a higher proliferative potential than CD138+ plasma cells, are CD138- [61], and hence not included in the positively selected plasma cells used in our study. In fact, current consensus recommendations include CD38, CD138, and CD45 as the best combination of backbone markers for the identification and selection of plasma cells; the addition of CD19 enables efficient distinction between normal/reactive and clonal plasma cell compartments based on their most frequent aberrant phenotypes [62,63,64]. However, multi-marker immunomagnetic enrichment of malignant plasma cells from BMA samples would be very laborious and most probably result in an insufficient cell number [65]. Cell selection based on this antibody panel for immunophenotyping could properly be performed by flow cytometry [66]; however, the amount of total RNA extracted by sorted malignant cells would be a limiting factor by itself for our experiments.

Summarizing the key findings of this work, we proved the abundance of tRFs in CD138+ plasma cells of MM and sMM patients. Moreover, we showed that some of these molecules represent putative molecular biomarkers that could distinguish between MM and sMM or between patients with and without osteolytic lesions. Regarding MM patients’ prognosis, all the investigated tRFs correlated with MM patients’ prognosis as analytically described above. All these results merit validation in a larger cohort. Additionally, the in silico functional analysis carried out for 3′-tRF-Leu^AAG/TAG^ revealed putative new aspects in the regulation of the MM molecular background.

Our future goals include the deciphering of the molecular mechanisms underlying the function of these tRFs in MM. This could reveal novel therapeutic strategies concerning MM treatment; such an attempt has been also suggested for miRNAs, but there are still important limitations, including impaired miRNA stability, side effects, and incapability of targeted miRNA delivery. These problems will be probably faced if similar attempts are made with tRFs. Additionally, the detection of these molecules in MM-derived EVs could contribute to our deeper understanding of their involvement in MM.

## Figures and Tables

**Figure 1 biomedicines-09-01811-f001:**
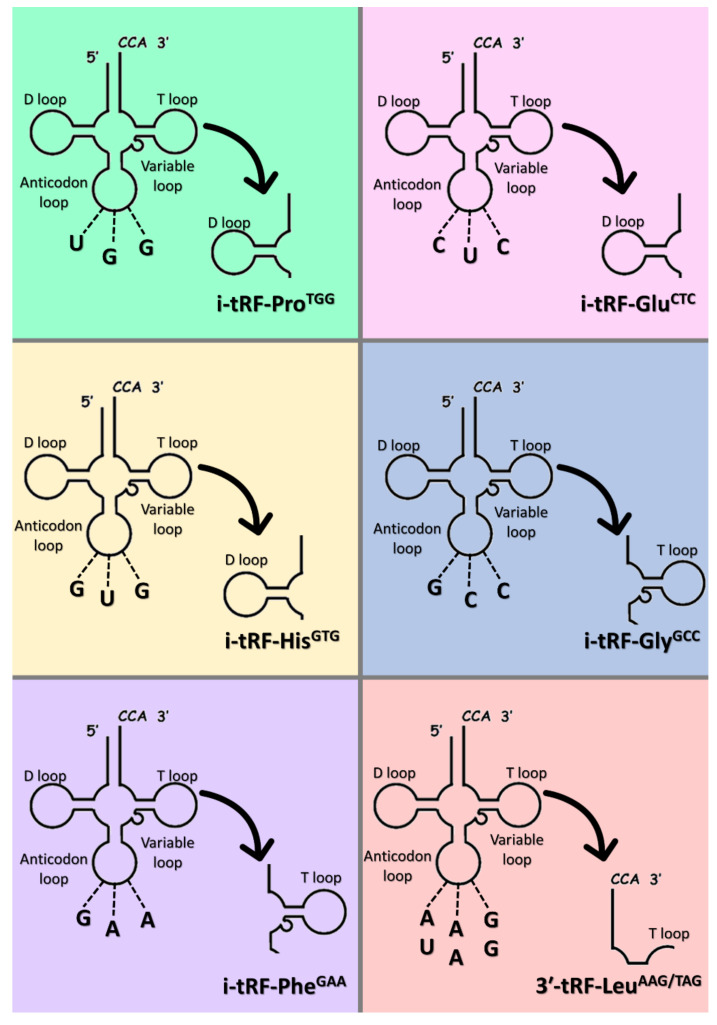
Graphical illustration of the mapping of the tRNA-derived RNA fragments (tRFs) investigated in this study with their respective tRNAs of origin.

**Figure 2 biomedicines-09-01811-f002:**
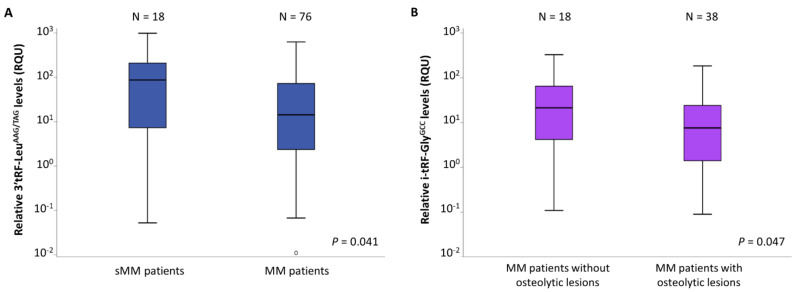
Boxplots, showing the differences of relative 3′-tRF-Leu^AAG/TAG^ levels between multiple myeloma (MM) and smoldering MM (sMM) patients (**A**)**,** and of relative i-tRF-Gly^GCC^ levels between MM patients without and with osteolytic lesions (**B**).

**Figure 3 biomedicines-09-01811-f003:**
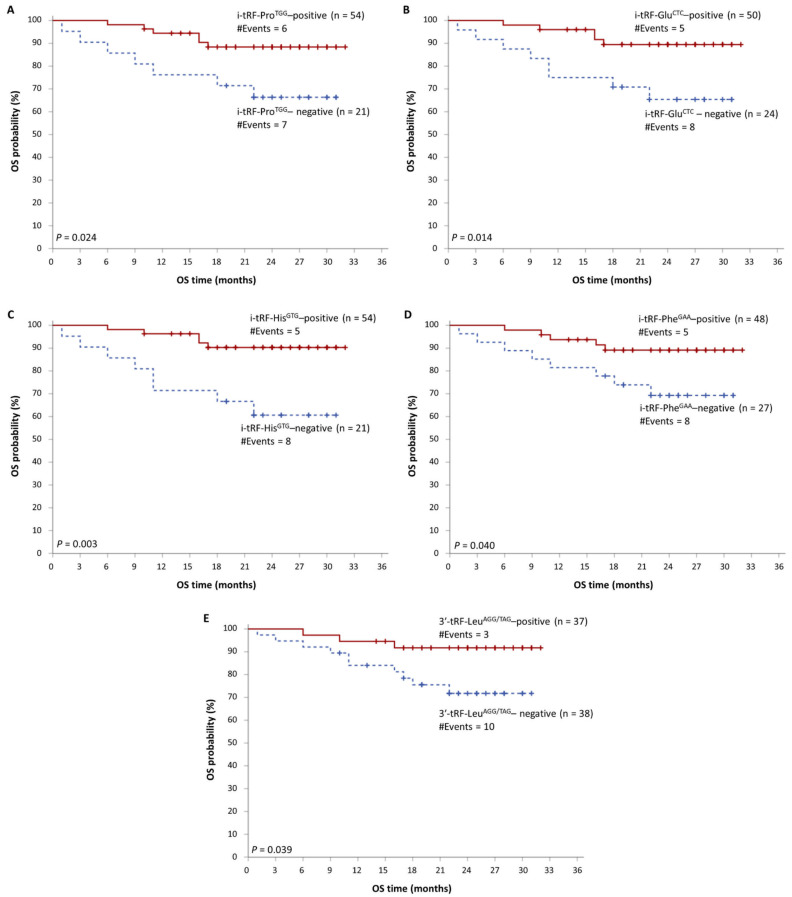
Kaplan–Meier overall survival (OS) curves, showing the differences in the survival intervals of MM patients with high levels of i-tRF-Pro^TGG^ (**A**), i-tRF-Glu^CTC^ (**B**), i-tRF-His^GTG^ (**C**), i-tRF-Phe^GAA^ (**D**), and 3′-tRF-Leu^AAG/TAG^ (**E**), compared to patients with lower levels of these molecules.

**Figure 4 biomedicines-09-01811-f004:**
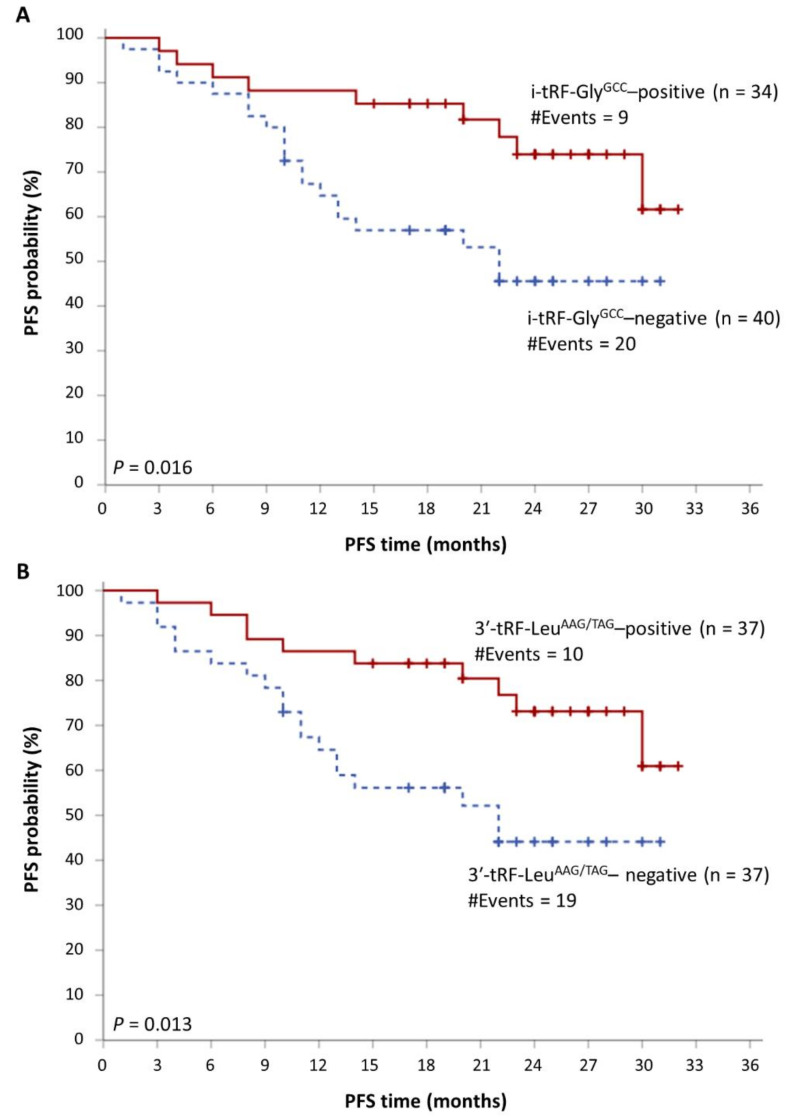
Kaplan–Meier progression-free survival (PFS) curves, showing the differences in the survival intervals of MM patients with high levels of i-tRF-Gly^GCC^ (**A**), and 3′-tRF-Leu^AAG/TAG^ (**B**), compared to patients with lower levels of these molecules.

**Figure 5 biomedicines-09-01811-f005:**
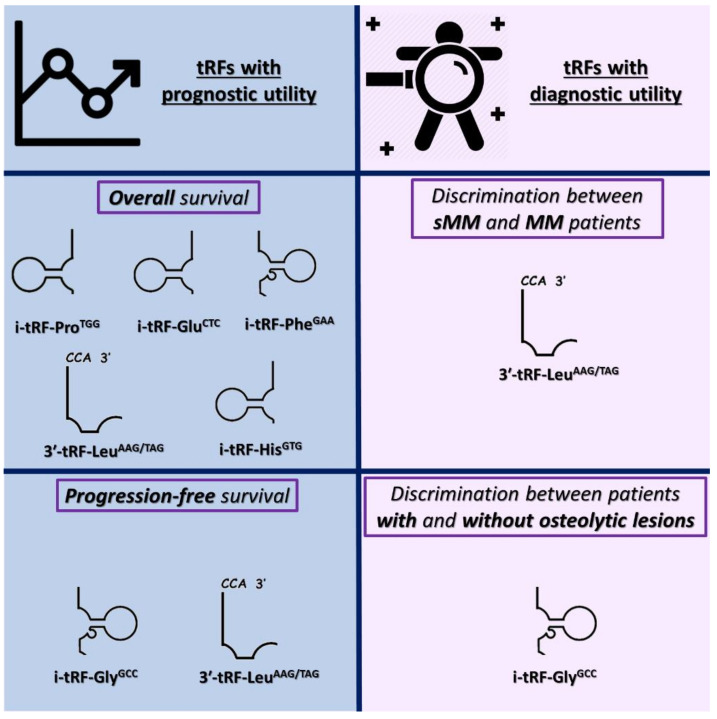
Graphical summary of the putative molecular biomarker utility of each tRF.

**Figure 6 biomedicines-09-01811-f006:**
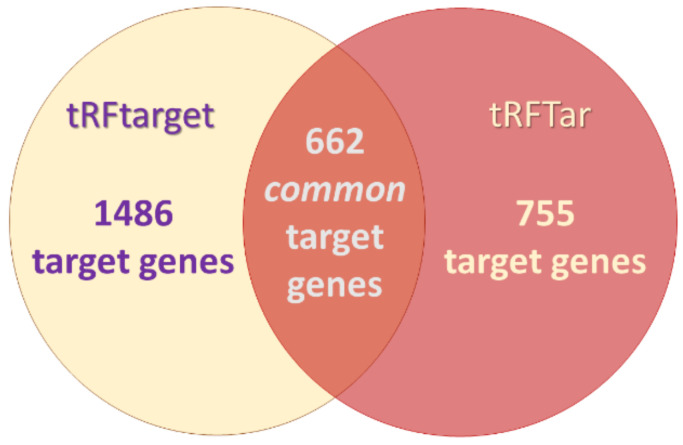
Venn chart showing the number of putative 3′-tRF-LeuAAG/TAG targets obtained from each database and their intersection, which was used for the Gene Ontology (GO) analysis.

**Figure 7 biomedicines-09-01811-f007:**
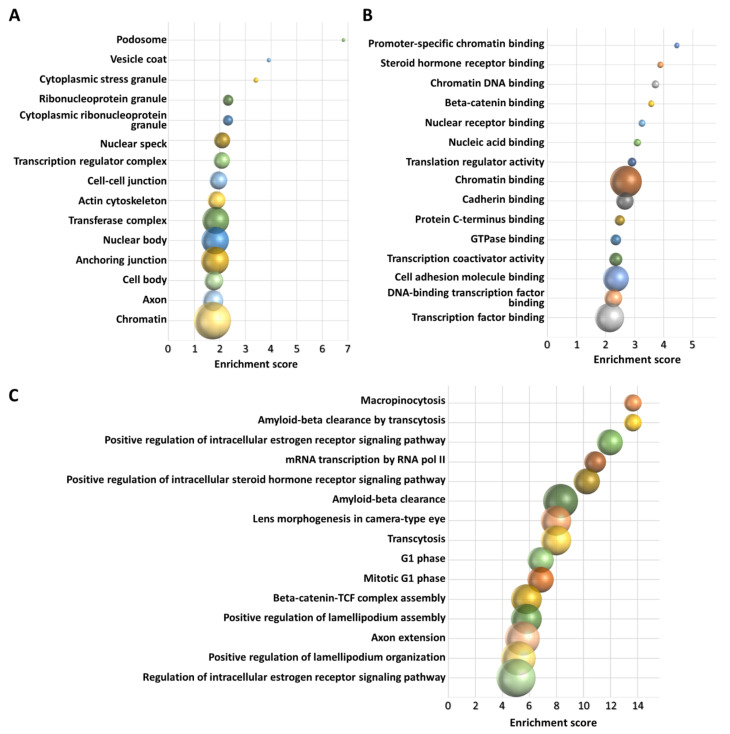
The results of functional GO analysis for 3′-tRF-Leu^AAG/TAG^. The cellular components (**A**), molecular functions (**B**), and biological processes (**C**) showing the highest enrichment scores are shown in the charts. The size of each bubble indicates the number of genes implicated. Each chart is drawn in scale.

**Table 1 biomedicines-09-01811-t001:** Characteristics of the multiple myeloma (MM) patients.

Variable	Number of Patients (%)
**Gender (76/76 patients)**	
Male	44 (57.9%)
Female	32 (42.1%)
**M-protein isotype (75/76 patients)**	
IgG	44 (58.7%)
IgA	17 (22.7%)
IgD	2 (2.7%)
Kappa light chain	7 (9.2%)
Lambda light chain	3 (4.0%)
Not typed	2 (2.7%)
**del(17p) (71/76 patients)**	
Absence	59 (83.1%)
Presence	12 (16.9%)
**t(4;14) (70/76 patients)**	
Absence	62 (88.6%)
Presence	8 (11.4%)
**t(14;16) (68/76 patients)**	
Absence	67 (98.5%)
Presence	1 (1.5%)
**(+1q) (54/76 patients)**	
Absence	30 (55.6%)
Presence	24 (44.4%)
**ISS ^1^ stage (74/76 patients)**	
I	15 (20.3%)
II	25 (33.8%)
III	34 (45.9%)
**R-ISS ^2^ stage (69/76 patients)**	
I	11 (15.9%)
II	40 (58.0%)
III	18 (26.1%)
**Bone disease (72/76 patients)**	
No	22 (30.6%)
Yes	50 (69.4%)
**WBLDCT ^3^ osteolysis (56/76 patients)**	
No	18 (32.1%)
Yes	38 (67.9%)

^1^ International Staging System; ^2^ Revised International Staging System; ^3^ Whole-body low-dose computed tomography.

**Table 2 biomedicines-09-01811-t002:** The tRNA-derived RNA fragments (tRFs) investigated in this study.

tRF	Fragment Sequence	Anticodon	Localization	Accession Number	MINTbase Unique ID
i-tRF-Pro^TGG^	5′-GUUGGUCUAGGGGUAUGAUUCUCGG-3′	UGG	Nucleus	MK671729	tRF-25-78WPRLXN48
i-tRF-Glu^CTC^	5′-GUCUAGUGGUUAGGAUUCGGCG-3′	CUC	Nucleus	MK671728	tRF-22-SX73V2Y8K
i-tRF-His^GTG^	5′-UGAUCGUAUAGUGGUUAGUACUCUGCG-3′	GUG	Nucleus	MW650833	tRF-27-XMSL73VL4YK
i-tRF-Gly^GCC^	5′-GAGGCCCGGGUUCGAUUC-3′	GCC	Nucleus	MK642309	tRF-18-5J3KYU05
i-tRF-Phe^GAA^	5′-UUUAGACGGGCUCACAUCACC-3′	GAA	Mitochondrion	MK671731	tRF-21-ZPEK45H5D
3’-tRF-Leu^AAG^^/^^TAG^	5′-AUCCCACCGCUGCCACCA-3′	AAG, UAG	Nucleus	MK671733	tRF-18-HR0VX6D2

**Table 3 biomedicines-09-01811-t003:** Primers used in real-time quantitative PCR (qPCR) for the relative quantification of the tRFs in all samples.

Amplified Molecule	Primer Sequence (5′→3′)	Direction	Length (nt ^1^)	T_m_ (°C)
i-tRF-Pro^TGG^	GTTGGTCTAGGGGTATGATTCTCGGA	Forward	26	62
i-tRF-Glu^CTC^	GTCTAGTGGTTAGGATTCGGCGA		23	61
i-tRF-His^GTG^	TGATCGTATAGTGGTTAGTACTCTGCG		27	59
i-tRF-Gly^GCC^	GAGGCCCGGGTTCGATTC		18	62
i-tRF-Phe^GAA^	TTTAGACGGGCTCACATCACC		21	59
3’-tRF-Leu^AAG^^/^^TAG^	ATCCCACCGCTGCCACCA		18	66
*SNORD43*	ACTTATTGACGGGCGGACA		19	59
*SNORD48*	TGATGATGACCCCAGGTAACTCT		23	59
Universal reverse	GCGAGCACAGAATTAATACGAC	Reverse	22	56

^1^ Nucleotides.

**Table 4 biomedicines-09-01811-t004:** Multivariate Cox regression analysis, regarding MM patients overall and progression-free survival.

	Covariate	HR ^1^	95% CI ^2^	*p* Value ^3^	BCa ^4^ Bootstrap ^5^ 95% CI ^2^	Bootstrap ^5^ *p* Value ^3^
**Overall survival (OS)**	**i-tRF-Pro^TGG^ status**					
	Positive	1.00				
	Negative	4.06	1.28–12.82	*0.0* *17*	0.98–46.70	*0.011*
	**R-ISS ^6^ (ordinal)**	3.39	1.28–8.96	*0.0* *14*	0.93–38.24	*0.024*
	**i-tRF-Glu^CTC^ status**					
	Positive	1.00				
	Negative	5.87	1.75–19.63	*0.0* *04*	1.02–5.58 × 10^5^	*0.001*
	**R-ISS ^6^ (ordinal)**	3.98	1.50–10.57	*0.006*	0.58 –7.39 × 10^5^	*0.010*
	**i-tRF-His^GTG^ status**					
	Positive	1.00				
	Negative	6.49	1.94–21.74	*0.0* *02*	0.97–7.32 × 10^5^	*0.001*
	**R-ISS ^6^ (ordinal)**	3.77	1.44–9.91	*0.0* *07*	0.87–4.42 × 10^5^	*0.008*
**Progression-free survival (PFS)**	**i-tRF-Gly^GCC^ status**					
	Positive	1.00				
	Negative	3.06	1.33–7.00	*0.008*	1.12–12.63	*0.0* *07*
	**ISS ^7^ (ordinal)**	2.22	1.25–3.95	*0.0* *07*	1.20–7.71	*0.008*
	**3′-tRF-Leu^AAG/TAG^ status**					
	Positive	1.00				
	Negative	2.94	1.32–6.55	*0.008*	1.28–8.76	*0.005*
	**ISS ^7^ (ordinal)**	2.16	1.22–3.80	*0.008*	1.21–6.37	*0.008*

^1^ Hazard ratio; ^2^ Confidence interval; ^3^ Italics indicate a significant *p* value; ^4^ Bias-corrected and accelerated; ^5^ Based on 1000 bootstrap samples; ^6^ Revised International Staging System; ^7^ International Staging System.

## Data Availability

The data presented in this study are available on request from the corresponding author. The data are not publicly available due to ethical issues.

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
