# Peer review of "tRNA Derivatives in Multiple Myeloma: Investigation of the Potential Value of a tRNA-Derived Molecular Signature"

_biomedicines, 2021, doi:10.3390/biomedicines9121811_

Round 1

Reviewer 1 Report

The authors analyzed for the first time in myeloma, they have previous experience in chronic lymphocytic leukemia, the role of a subset of non-coding RNA, specifically tRNA derived RNA fragments (tRFs). They studied their expression in the medullary plasmocytes of SMM and MM. Six of these tRFs had an independent prognostic role (PFS and/or OS) and for one of them they were able to conduct functional analysis. They could also discriminate between SMM and MM as well as myeloma with or without bone lesions.

They claim that it has never been reported before. The paper is well written.

Minor revision:

Figure 5: the Y axis should be PFS and not cumulative survival probability

The authors analyzed for the first time in myeloma, they have previous experience in chronic lymphocytic leukemia, the role of a subset of non-coding RNA, specifically tRNA derived RNA fragments (tRFs). They studied their expression in the medullary plasmocytes of SMM and MM. Six of these tRFs had an independent prognostic role (PFS and/or OS) and for one of them they were able to conduct functional analysis. They could also discriminate between SMM and MM as well as myeloma with or without bone lesions.

They claim that it has never been reported before. The paper is well written.

Minor revision:

Figure 5: the Y axis should be PFS and not cumulative survival probability

Author Response

Figure 5: the Y axis should be PFS and not cumulative survival probability

We thank the Reviewer for his positive appraisal of our original research article. In response to his remark, we replaced “cumulative survival probability” with “OS probability” or “PFS probability” where appropriate, in all Figures and Supplementary Material.

The Authors wish to thank the Reviewers for their constructive comments that led to the improvement of the current manuscript.

Reviewer 2 Report

This is an excellent article that highlights the potential use of tRFs as biomarkers that are predictive of prognosis and presence of osteolytic lesions in MM patients. The data is well-presented and explains how the level of tRFs can predict survival, progression, and osteolytic lesions. There is also a functional section that explores the potential mechanism of action linking one tRF to MM progression. More work is required to link these tRF changes to pathophysiology of MM, but the research methods are still catching up. I think the authors highlight well the limitations of their study including small sample size and lack of mechanistic studies. Overall, this is a strong paper that is worthy of publication as it will spur on future studies of diagnostic and mechanistic value to the field. 

Author Response

This is an excellent article that highlights the potential use of tRFs as biomarkers that are predictive of prognosis and presence of osteolytic lesions in MM patients. The data is well-presented and explains how the level of tRFs can predict survival, progression, and osteolytic lesions. There is also a functional section that explores the potential mechanism of action linking one tRF to MM progression. More work is required to link these tRF changes to pathophysiology of MM, but the research methods are still catching up. I think the authors highlight well the limitations of their study including small sample size and lack of mechanistic studies. Overall, this is a strong paper that is worthy of publication as it will spur on future studies of diagnostic and mechanistic value to the field.

We thank the Reviewer for his positive appraisal of our original research article.

The Authors wish to thank the Reviewers for their constructive comments that led to the improvement of the current manuscript.

Reviewer 3 Report

Karousi et al. introduced a clinical study involving MM patients where they looked at the possible utilization of tRFs as diagnostic and prognostic biomarkers. A 24 month follow-up provided a robust insight into the use of these tRFs in a natural MM environment as prognostic markers through assessing overall and progression-free survival. The field of characterizing the expression of small non-coding RNA fragments such tRFs, especially in MM, is still new. This added novelty and scientific significance to the study. However, there are some issues, especially with study design should be addressed as follows: 

1) Authors repeatedly mentioned that this is "the first study to uncover the potential clinical significance of such fragments in MM" in the abstract, introduction and discussion sections. However, previously published papers have looked into tRFs and their role in MM. For example: 

https://dx.doi.org/10.2147%2FOTT.S302594

https://doi.org/10.1186/s12859-021-04167-8

https://doi.org/10.1158/1538-7445.AM2021-2394

 So please acknowledge their contribution to the field and cite these articles in the text.

2) Also, the authors have published a work in 2019 that involved tRFs and MM (https://doi.org/10.1182/blood-2019-129082) where they have mentioned similar data reported in this manuscript. 

For example, authors reported that 3′-tRF-LeuAAG/TAG expression levels differed significantly between MM and SMM cases which is similar to the findings of this study. 

So please explain if there is a difference between the two studies or remove any repeated data from the manuscript

3) Experimental design: a control group represented by healthy individuals/donors should be added to all data sets to define a baseline to confirm that all changes in tRFs are correlated with MM and SMM. This will also support the value of using these molecules as diagnostic biomarkers to differentiate between healthy non-MM and MM patients as well as MM and SMM.

4) A rationale for specifically isolating CD138+ plasma cells for RNA extraction is not clear

5) Did the authors evaluate RNA quality in addition to quantification using for example, Bioanalyzer?

6) Please move table of primers sequences from supplementary material into text

7) Please check the numbering of Results section subheadings

8)  3. Results 3.1. Development of a qPCR assay. Would you please move this part of the results to methods section and Figure 2 to supplementary material  

9) 3.1.1.3.23′-. tRF-LeuAAG/TAG levels are significantly higher in SMM patients’ plasma cells, compared to plasma cells from MM patients.

Authors previously reported that the levels of 3′-tRF-LeuAAG/TAG, itRF-GluCTC, and i-tRF-ProTGG differed significantly between MM and SMM cases (P=0.001, P=0.047 and P=0.033, respectively) in a recent study: https://doi.org/10.1182/blood-2019-129082. Though, in the current study, authors reported that only tRF-LeuAAG/TAG levels are significantly higher in SMM patients’ plasma cells, compared to plasma cells from MM patients.

So, please explain the difference between the two studies and why not all three tRFs were differentially expressed in the current study.  

Author Response

 Authors repeatedly mentioned that this is "the first study to uncover the potential clinical significance of such fragments in MM" in the abstract, introduction, and discussion sections. However, previously published papers have looked into tRFs and their role in MM. For example: 

  • https://dx.doi.org/10.2147%2FOTT.S302594
  • https://doi.org/10.1186/s12859-021-04167-8
  • https://doi.org/10.1158/1538-7445.AM2021-2394

So please acknowledge their contribution to the field and cite these articles in the text.

We thank the Reviewer for this remark. We have deleted the term “the first study” everywhere in the manuscript. Moreover, the Introduction has been accordingly modified to include the suggested, previously published papers.

Page 2  (Lines 86-87): The present study aimed to further uncover the molecular biomarker utility of tRFs in MM, as only a few studies have aimed to elucidate their role in this disease so far [28-30].

Moreover, we added the appropriate references, as aforementioned:

  1. Xu, C.; Fu, Y. Expression Profiles of tRNA-Derived Fragments and Their Potential Roles in Multiple Myeloma. Onco Targets Ther 2021, 14, 2805-2814, doi:10.2147/ott.S302594.
  2. Rojek, A.E.; Katanski, C.D.; Stefka, A.; Derman, B.A.; Jakubowiak, A.; Pan, T. Abstract 2394: tRNA expression and tRFs in multiple myeloma: Progression from monoclonal gammopathies to relapsed/refractory disease. Cancer Research 2021, 81, 2394-2394, doi:10.1158/1538-7445.Am2021-2394.
  3. Xu, C.; Liang, T.; Zhang, F.; Liu, J.; Fu, Y. tRNA-derived fragments as novel potential biomarkers for relapsed/refractory multiple myeloma. BMC Bioinformatics 2021, 22, 238, doi:10.1186/s12859-021-04167-8.

  1. Also, the authors have published a work in 2019 that involved tRFs and MM (https://doi.org/10.1182/blood-2019-129082) where they have mentioned similar data reported in this manuscript. For example, authors reported that 3′-tRF-LeuAAG/TAG expression levels differed significantly between MM and SMM cases which is similar to the findings of this study. So please explain if there is a difference between the two studies or remove any repeated data from the manuscript.

This work mentioned by the Reviewer includes previous preliminary results of the current study, which had been announced to the 61st American Society of Hematology (ASH 2019) Annual Meeting, and have thus been included in this Supplement of Blood journal. At that time, we had no survival data yet and focused on the discriminatory value of three tRNA-derived RNA fragments. Moreover, those preliminary results were based on a smaller size of both cohorts of MM and sMM patients. Therefore, there is no overlap between that abstract and the current manuscript.

  1. Experimental design: a control group represented by healthy individuals/donors should be added to all data sets to define a baseline to confirm that all changes in tRFs are correlated with MM and SMM. This will also support the value of using these molecules as diagnostic biomarkers to differentiate between healthy non-MM and MM patients as well as MM and SMM.

We agree with the Reviewer. However, this study is based on bone marrow asprirate (BMA) samples, it is impossible to include BMA samples from healthy blood donors (normal controls). A cohort consisting of healthy individuals could not be available, since BM is a painful procedure, performed only under specific circumstances where the presence of particular hematological diseases is suspected.

  1. A rationale for specifically isolating CD138+ plasma cells for RNA extraction is not clear.

We thank the Reviewer for this important comment. We added a few sentences in the  Materials and Methods section, to explain the rationale for specifically isolating CD138+ plasma cells for RNA extraction , as also explained below:

Page 3, (Lines 122-126): In order to select plasma cells from BMA samples of MM and SMM patients, we performed CD138-positive selection among mononuclear cells of BMA as a method of choice.  CD138, officially known as Syndecan 1, is a transmembrane (type I) heparan sulfate proteoglycan encoded by the human SDC1 gene; its expression is considered as a hallmark of MM cells and plasma cells in the bone marrow [31-33].

We also added the appropriate references:

  1. Ridley, R.C.; Xiao, H.; Hata, H.; Woodliff, J.; Epstein, J.; Sanderson, R.D. Expression of syndecan regulates human myeloma plasma cell adhesion to type I collagen. Blood 1993, 81, 767-774.
  2. Kawano, Y.; Fujiwara, S.; Wada, N.; Izaki, M.; Yuki, H.; Okuno, Y.; Iyama, K.; Yamasaki, H.; Sakai, A.; Mitsuya, H., et al. Multiple myeloma cells expressing low levels of CD138 have an immature phenotype and reduced sensitivity to lenalidomide. Int J Oncol 2012, 41, 876-884, doi:10.3892/ijo.2012.1545.
  3. Wijdenes, J.; Vooijs, W.C.; Clement, C.; Post, J.; Morard, F.; Vita, N.; Laurent, P.; Sun, R.X.; Klein, B.; Dore, J.M. A plasmocyte selective monoclonal antibody (B-B4) recognizes syndecan-1. Br J Haematol 1996, 94, 318-323, doi:10.1046/j.1365-2141.1996.d01-1811.x.

  1. Did the authors evaluate RNA quality in addition to quantification using for example, Bioanalyzer?

We thank the Reviewer for this question. The RNA integrity was assessed by agarose gel electrophoresis, prior to in vitro polyadenylation. We clarified this in the Materials and Methods section:

Page 4 (Lines 134-136): The concentration of each total RNA was determined using Qubit™ 2 Fluorometer (Invitrogen™, Thermo Fisher Scientific Inc., Carlsbad, CA, USA), and its integrity was assessed by agarose gel electrophoresis.

  1. Please move table of primers sequences from supplementary material into text.

We complied with the Reviewer’s suggestion; this is Table 2 (Page 6) in the revised manuscript.

  1. Please check the numbering of Results section subheadings.

We fixed this issue.

  1. Results 3.1. Development of a qPCR assay. Would you please move this part of the results to methods section and Figure 2 to supplementary material?

We followed the Reviewer’s advice. Thus, me modified the Materials and Methods section, and moved previous Figure 2 to the Supplementary Material (Figure S1 in the revised manuscript).

Page 6 (Lines 183-185): The qPCR assay was optimized, in order to observe a unique melting curve for each amplicon, and thus be able to quantify the selected tRFs in all samples. The melting curves for each tRF, as well as those of the reference molecules, are shown in Figure S1.

  1. Authors previously reported that the levels of 3′-tRF-LeuAAG/TAG, itRF-GluCTC, and i-tRF-ProTGG differed significantly between MM and SMM cases (P=0.001, P=0.047 and P=0.033, respectively) in a recent study: https://doi.org/10.1182/blood-2019-129082. Though, in the current study, authors reported that only tRF-LeuAAG/TAG levels are significantly higher in SMM patients’ plasma cells, compared to plasma cells from MM patients. So, please explain the difference between the two studies and why not all three tRFs were differentially expressed in the current study.

As explained above, this work mentioned by the Reviewer includes previous preliminary results of the current study, which had been announced to the 61st American Society of Hematology (ASH 2019) Annual Meeting, and have thus been included in this Supplement of Blood journal. At that time, the size of both cohorts (MM and sMM patients) was smaller; thus, the levels of two of the studied tRFs (itRF-GluCTC and i-tRF-ProTGG) differed significantly between these two cohorts – as correctly mentioned by the Reviewer – yet this statistical significance was marginal at that time. Now, using a little bit larger cohorts, both these differences show a P value that is slightly >0.050 and hence are considered as rather insignificant. For this reason, we omitted any reference to them.

The Authors wish to thank the Reviewers for their constructive comments that led to the improvement of the current manuscript.

Reviewer 4 Report

This is an interesting study which is very well presented. The use of figures in the main document and supplemental material is excellent. However, there are  some questions I would have about the patient selection, a few aspects of the methodology and importantly, a tendency to over interpret some of the findings, and the overall significance of the study in real world haematology practice. These include:

  • A median patient follow-up of 24 months is much too short to draw definitive conclusions.
  • The patient cohort contained only 28% of patients who underwent ASCT which seems low in a real world situation.
  • If I am interpreting the data correctly, patients received either a bortezomib based drug regimen OR an IMiD based regimen (was that thalidomide, lenalidomide or other?). Again, nowadays it is quite  common for patients to be given a combination of a proteasome inhibitor and an immunomodulatory drug as induction therapy, so the treatment of this patient cohort may not be representative of practice in many centres. 
  • I found little data on existing predictive biomarkers and whether these showed any correlation with the 5 tRFs identified as predictors of better outcomes. For example, was there an absence of chromosomal aberrations associated with poorer outcomes, such as del17p, in the patients predicted by tRF analyses to have better OS? Also was the M-protein isotype correlated in any way with the tRF levels?
  • I would have been particularly interested in the results of the two IgD patients. As the authors will know, these IgD patients typically present at a younger age and have shorter OS that the common MM isotopes. Also, there is some evidence they have low IL-6 levels at presentation, of interest given the references to IL-6 in this paper.
  • CD138 selection is not the best method for selection of tumour plasma cells. Syndecan 1 is present in normal as well as malignant plasma cells. This becomes important when there are significant numbers of normal plasma cells present, as may be the case in MGUS, SMM and MM patients who have been diagnosed early in the disease pathway. Although not often done, it is better to use a combination of mAb to identify and select the tumour cells.

In the penultimate and final paragraphs the authors acknowledge the limitations of their study and outline their future goals. These paragraphs are excellent and it will be interesting to see if the preliminary predictions of increased OS, or PFS, hold up. My personal view is that the greatest value of this work may be the potential to reveal novel therapeutic targets and I would encourage this group to pursue their studies.

Author Response

  1. A median patient follow-up of 24 months is much too short to draw definitive conclusions.

We totally agree with the Reviewer that the rather short follow-up time is a limitation of our study. Therefore, we discuss this in our manuscript:

Page 15 (Lines 415-419): Our study has some limitations. […] Secondly, the median follow-up time of the patients included in the current study is only 2 years; however, a longer follow-up time period is needed to uncover the emerging potential of particular tRFs as important molecular biomarkers in MM.

  1. The patient cohort contained only 28% of patients who underwent ASCT which seems low in a real-world situation.

Prompted by the Reviewer’s remark, we explained in the Materials and Methods of the revised manuscript the reasons for which only 27.6% of the MM patients of our study were subjected to HDM-ASCT, as presented here below:

Page 3 (Lines 113-117): Moreover, 21 (27.6%) out of 76 MM patients were subjected to autologous stem cell transplantation following high-dose melphalan, whereas the rest 55 (72.4%) MM patients were not candidates for bone marrow transplant, either because of being older than 65 years (48 cases) or because of severe comorbidities and/or impaired performance status (7 cases).

  1. If I am interpreting the data correctly, patients received either a bortezomib-based drug regimen OR an IMiD based regimen (was that thalidomide, lenalidomide or other?). Again, nowadays it is quite common for patients to be given a combination of a proteasome inhibitor and an immunomodulatory drug as induction therapy, so the treatment of this patient cohort may not be representative of practice in many centres. 

We thank the Reviewer for this comment, which led us to clarify this very important issue. In the Materials and Methods of the revised manuscript, we have included a detailed description of the drug-based therapy regimen that each MM patient received:

Page 3 (Lines 108-113): Only one patient did not receive any treatment, while the treatment of the rest 75 patients varied. More specidically, 63 out of the 75 MM patients (84.0%) were treated with bortezomib plus an immunomodulatory drug [either lenalidomide (60 cases) or thalidomide (3 cases)], 6 (8.0%) MM patients received bortezomib along with cyclophosphamide and dexamethasone, and 6 (8.0%) MM patients were treated with le-nalidomide and dexamethasone.

  1. I found little data on existing predictive biomarkers and whether these showed any correlation with the 5 tRFs identified as predictors of better outcomes. For example, was there an absence of chromosomal aberrations associated with poorer outcomes, such as del17p, in the patients predicted by tRF analyses to have better OS? Also was the M-protein isotype correlated in any way with the tRF levels?

With respect to the Reviewer’s remark, we added these results in the revised manuscript:

Page 8 (Lines 240-246): MM patients were subgrouped based on the levels of each tRF, based on the cut-off points determined for prognostic purposes using the X-tile software, as aforementioned. The frequencies of MM patient subgroups are shown in Table S1. Inrestingly, positive i-tRF-GlyGCC status was found to be associated with the presence of chromosomal aberration (+1q) (Table S2). Associations between each tRF status and del(17p), t(4;14) or t(14;16) could not be checked, due to the limited number of MM patients bearing at least one of these chromosomal aberrations.

Page 7 (Lines 226-230): In order to examine potential relationships between tRFs and MM features, we checked whether the tRF levels differed between dinstict subroups of MM patients, including those with different M-protein isotype as well as patients with or without osteolytic lesions. No statistically significant differences were observed in the levels of each tRF among MM patients with different M-protein isotype (IgG and IgA).

Moreover, we discussed these findings:

Pages 15-16 (Lines 417-423): Due to the small sample size, our study can only present some evidence about the clin-ical utility of the aforementioned tRFs as molecular biomarkers in MM. For the same reason, no associations between tRF levels and features of patients with rare MM types, such IgD and non-secretory myeloma type, could be studied. Moreover, the re-sults regarding the association of positive i-tRF-GlyGCC status with the presence of (+1q) need to be validated in a larger cohort of MM patients.

  1. I would have been particularly interested in the results of the two IgD patients. As the authors will know, these IgD patients typically present at a younger age and have shorter OS than the common MM isotopes. Also, there is some evidence they have low IL-6 levels at presentation, of interest given the references to IL-6 in this paper.

Immunoglobulin D (IgD) myeloma is a rare isotype that comprises 1–2% of MM patients, which has significantly inferior survival for a median overall survival (OS) between 13 and 21 months. Therefore, we agree with the Reviewer that IgD patients are of special interest; indeed, in our cohort, one of them showed a short OS time (6 months), while the other one exhibited disease progression. However, the number is too small to draw any conclusion. On the other hand, excluding the 2 IgD MM patients from the cohort would produce a small bias; this is why we chose not to exclude any myeloma type from the MM patient cohort.

Additionally, we decided to add a couple of sentences in the Discussion section, in which we discussion the limitations of our study, admitting that our study does not allow conclusions to be drawn regarding the IgD subgroup of MM patients:

Page 15 (Lines 417-421): Due to the small sample size, our study can only present some evidence about the clinical utility of the aforementioned tRFs as molecular biomarkers in MM. For the same reason, no associations between tRF levels and features of patients with rare MM types, such IgD and non-secretory myeloma type, could be studied.

  1. CD138 selection is not the best method for selection of tumour plasma cells. Syndecan 1 is present in normal as well as malignant plasma cells. This becomes important when there are significant numbers of normal plasma cells present, as may be the case in MGUS, SMM and MM patients who have been diagnosed early in the disease pathway. Although not often done, it is better to use a combination of mAb to identify and select the tumour cells.

We thank very much the Reviewer for this comment, which definitely resulted in the improvement of our manuscript. In the revised version of the manuscript, we clearly set this as a limitation, yet we explain meanwhile that this is an inherent limitation that could not be overcome:

Page 16 (Lines 428-442): Lastly, another limitation is that the selected CD138+ plasma cells included not only ma-lignant plasma cells but also normal ones [62]. Furthermore, a subpopulation of malig-nant plasma cells, which are more primitive and which have a higher proliferative poten-tial than CD138+ plasma cells, are CD138- [63], and hence not included in the positively selected plasma cells used in our study. In fact, current consensus recommendations include CD38, CD138, and CD45 as the best combination of backbone markers for the iden-tification and selection of plasma cells; the addition of CD19 enables efficient distinction between normal/reactive and clonal plasma cell compartments based on their most fre-quent aberrant phenotypes [64-66]. However, multi-marker immunomagnetic enrichment of malignant plasma cells from BMA samples would be very laborious and most probably result in an insuficient cell number [67]. Cell selection based on this antibody panel for immunophenotyping could properly be performed by flow cytometry [68]; however, the amount of total RNA extracted by sorted malignant cells would be a limiting factor by itself for our experiments.

We also added the respective references:

  1. Kumar, S.; Kimlinger, T.; Morice, W. Immunophenotyping in multiple myeloma and related plasma cell disorders. Best Pract Res Clin Haematol 2010, 23, 433-451, doi:10.1016/j.beha.2010.09.002.
  2. Reid, S.; Yang, S.; Brown, R.; Kabani, K.; Aklilu, E.; Ho, P.J.; Woodland, N.; Joshua, D. Characterisation and relevance of CD138-negative plasma cells in plasma cell myeloma. Int J Lab Hematol 2010, 32, e190-196, doi:10.1111/j.1751-553X.2010.01222.x.
  3. Harada, H.; Kawano, M.M.; Huang, N.; Harada, Y.; Iwato, K.; Tanabe, O.; Tanaka, H.; Sakai, A.; Asaoku, H.; Kuramoto, A. Phenotypic difference of normal plasma cells from mature myeloma cells. Blood 1993, 81, 2658-2663.
  4. Paiva, B.; Almeida, J.; Perez-Andres, M.; Mateo, G.; Lopez, A.; Rasillo, A.; Vidriales, M.B.; Lopez-Berges, M.C.; Miguel, J.F.; Orfao, A. Utility of flow cytometry immunophenotyping in multiple myeloma and other clonal plasma cell-related disorders. Cytometry B Clin Cytom 2010, 78, 239-252, doi:10.1002/cyto.b.20512.
  5. Rawstron, A.C.; Orfao, A.; Beksac, M.; Bezdickova, L.; Brooimans, R.A.; Bumbea, H.; Dalva, K.; Fuhler, G.; Gratama, J.; Hose, D., et al. Report of the European Myeloma Network on multiparametric flow cytometry in multiple myeloma and related disorders. Haematologica 2008, 93, 431-438, doi:10.3324/haematol.11080.
  6. Beasley, A.B.; Acheampong, E.; Lin, W.; Gray, E.S. Multi-Marker Immunomagnetic Enrichment of Circulating Melanoma Cells. Methods Mol Biol 2021, 2265, 213-222, doi:10.1007/978-1-0716-1205-7_16.
  7. van Dongen, J.J.; Lhermitte, L.; Böttcher, S.; Almeida, J.; van der Velden, V.H.; Flores-Montero, J.; Rawstron, A.; Asnafi, V.; Lécrevisse, Q.; Lucio, P., et al. EuroFlow antibody panels for standardized n-dimensional flow cytometric immunophenotyping of normal, reactive and malignant leukocytes. Leukemia 2012, 26, 1908-1975, doi:10.1038/leu.2012.120.

The Authors wish to thank the Reviewers for their constructive comments that led to the improvement of the current manuscript.

Round 2

Reviewer 3 Report

The authors have successfully cleared my concerns. One last suggested change. Since it is difficult to include BMA samples from healthy blood donors and the inability to have a cohort consisting of healthy individuals due to the painful BMA procedure, authors should highlight this in limitations and that missing these controls may affect the conclusions of correlation of tRFs expression and SMM/MM diagnosis or prognosis. 

Author Response

  1. The authors have successfully cleared my concerns. One last suggested change. Since it is difficult to include BMA samples from healthy blood donors and the inability to have a cohort consisting of healthy individuals due to the painful BMA procedure, authors should highlight this in limitations and that missing these controls may affect the conclusions of correlation of tRFs expression and SMM/MM diagnosis or prognosis. 

Following the Reviewer’s comment, we added the following limitation in the Discussion section:

Page 16 (Lines 426-431): Additionally, a cohort consisting of normal controls is not available, as BMA is a painful procedure with rare but still possible complications, performed only when the presence of particular hematological diseases are suspected. However, unprocessed bone marrow samples could be purchased and used for comparison purposes. Inclusion of such samples could strengthen our conclusions about the diagnostic and prognostic utility of these tRFs in MM.

The Authors wish to thank the Reviewer for their constructive comments that led to the improvement of the current manuscript.
